# Correlates of COVID-19 conspiracy theory beliefs in Japan: A cross-sectional study of 28,175 residents

Yukihiro Sato [1,2] *, Ichiro Kawachi[3], Yasuaki Saijo[1], Eiji Yoshioka[1], Ken Osaka[2], Takahiro Tabuchi [4,5,6]

1 Department of Social Medicine, Asahikawa Medical University, Asahikawa, Hokkaido, Japan,
2 Department of International and Community Oral Health, Tohoku University Graduate School of Dentistry, Sendai, Miyagi, Japan, 3 Department of Social and Behavioral Sciences, Harvard T.H. Chan School of Public Health, Boston, Massachusetts, United States of America, 4 Division of Epidemiology, School of Public Health, Tohoku University Graduate School of Medicine, Sendai, Miyagi, Japan, 5 Cancer Control Center, Osaka International Cancer Institute, Osaka, Osaka, Japan, 6 The Tokyo Foundation for Policy Research, Minato-ku, Tokyo, Japan

* ys@epid.work

**Data Availability Statement:** The data from this study are not stored in a public repository due to the inclusion of personally identifiable or sensitive information. In accordance with Japan's ethical

## Abstract

### Background

The COVID-19 pandemic was associated with an increase in conspiracy theories worldwide. However, in Japan, the prevalence of COVID-19 conspiracy beliefs has remained unclear. This study aimed to estimate the prevalence and correlates of COVID-19 conspiracy beliefs using a survey of 28,175 residents of Japan aged 16–81 years old.

### Methods

A cross-sectional self-administered survey was conducted from September to October 2021. To assess the number of COVID-19 conspiracy beliefs, we used three questions from the Oxford Coronavirus Explanations, Attitudes, and Narratives Survey. Independent variables included general vaccine conspiracy beliefs, sociodemographic variables, information sources for COVID-19, trust in authorities, and fear of COVID-19.

### Results

After applying sampling weights and imputation, the estimated prevalence of holding at least one COVID-19 conspiracy belief was 24.4%. From a linear regression model, several factors were independently associated with conspiracy beliefs. Notably, people with the lowest level of education (lower secondary school) endorsed fewer COVID-19 conspiracy beliefs (B -0.089, vs. upper secondary school). Furthermore, higher socioeconomic backgrounds–such as higher income, higher wealth, and regular employment–were associated with endorsing conspiracy beliefs. Only 37.3% of respondents trusted the government of Japan, but paradoxically, trust in the government was positively associated with conspiracy beliefs (B 0.175, vs. distrust).

guidelines, the Research Ethics Committee of the Osaka International Cancer Institute imposed restrictions on data sharing. For any inquiries regarding the data, please contact the JACSIS research office (jacsis_jastis_office@ymail.ne.jp). Details on data availability can be found on the JACSIS website (https://jacsis-study.jp/dug/index.html).

**Funding:** This study was funded by the Japan Society for the Promotion of Science (JSPS) KAKENHI Grants (grant numbers 18H03062). The funders had no role in study design, in the collection, analysis and interpretation of data, in the writing of the articles, and in the decision to submit it for publication.

**Competing interests:** The authors have declared that no competing interests exist.

## Conclusions

COVID-19 conspiracy beliefs can be prevalent in about a quarter of the residents of Japan. Certain groups are more likely to endorse conspiracy beliefs, and targeting interventions towards these groups might be efficient in stemming the spread of conspiracy beliefs.

## Introduction

The COVID-19 pandemic has presented challenges to global health, not only due to the virus itself, but also due to the rapid spread of misinformation, particularly conspiracy theories [1, 2]. Conspiracy theories, defined as "explanations attributing significant events to a small group of powerful actors engaging in officially unacknowledged malevolent activities" [3–7], involve alternative explanations for the emergence of the novel virus, e.g., that the virus was created to reduce the global population, or that pharmaceutical companies manufactured and spread the virus to reap profits [8]. Such COVID-19 conspiracy theories have undermined public health measures, such as social distancing, hand washing, and mask-wearing [9–13].

As in other countries, COVID-19 conspiracy theories have proliferated in Japan. In 2022, for example, a Trump-inspired political party (Sansei tou), which advanced conspiracy theories about COVID-19 (e.g. laboratory origin narrative and big Pharma plot narratives), secured a seat in the House of Councillors [14]. Conspiracy theories appear to be gaining popularity in Japan. However, in Japan, although a previous study estimated the prevalence of conspiracy theories regarding general vaccines [15], the prevalence of COVID-19 conspiracy theories remains unclear.

Previous studies examined the characteristics of individuals endorsing COVID-19 conspiracy beliefs [13]. Based on previous reports, potential determinants of harboring COVID-19 conspiracy beliefs include socioeconomic and social factors, while potential consequences include health behaviours such as vaccine hesitancy and disregard for hygiene guidelines. Among these antecedents, trust has emerged as a key factor influencing the acceptance or rejection of scientific consensus, particularly regarding conspiracy theories. Scientific information is not directly transmitted from scientists, but relayed by governments and national health institutions [12, 13]. Distrust in authorities hampers scientific communication. In addition, trust in science and scientists is also essential. An early study identified trust in science as the strongest predictor of rejecting COVID-19 conspiracy theories [16]. Strengthening trust in science may be a key strategy to counter harmful false beliefs.

Socioeconomic status is also a key factor in predicting conspiracy beliefs [12, 13]. Individuals with lower educational attainment were more likely to believe COVID-19 conspiracy theories, potentially due to factors such as intuitive thinking, a lower sense of control, and exposure to low-quality information [13, 17]. COVID-19 conspiracy theories are abundant on the internet, especially on social media platforms such as YouTube, Instagram, and Twitter (currently known as X). These platforms can amplify conspiracy narratives [1, 18] and be one source of conspiracy theories going viral in countries such as Japan [19]. An early study revealed that those relying more on social media (relative to mainstream media) for COVID-19 news had lower health literacy [20] and were more susceptible to misinformation and conspiracy theories [20, 21].

The endorsement of conspiracy theory can be contingent upon sociocultural factors [22], and Japan presents a unique context. Studies have found that individuals who rely on traditional media tend to be less inclined to believe in conspiracy theories [23, 24]. However, even

in traditional media such as TV news and newspapers, there is a history of biased health reporting in Japan. For example, Japan's human papillomavirus vaccination rate is low, due to excessive reports of false side effects by traditional media [25]. This situation might have facilitated distrust in government officials, healthcare professionals, and scientists. Indeed, Japan had relatively low trust in the government in 2020 among OECD countries [26]. The situation might have increased the prevalence of COVID-19 conspiracy beliefs. Therefore, this study aimed to estimate the prevalence of COVID-19 conspiracy beliefs and explore their correlates among 28,175 residents of Japan.

## Methods

### Ethics approval and consent to participate

The study protocol was reviewed and approved by the Research Ethics Committee of the Osaka International Cancer Institute (approved on June 19, 2020; approval number 20084). The authors assert that all procedures contributing to this work comply with the ethical standards of the relevant national and institutional committees on human experimentation and with the Helsinki Declaration of 1975, as revised in 2008. All participants provided written informed consent online before responding to the questionnaire. Participants received Rakuten points as a reward, which can be used for online shopping. The exact number of points is not disclosed at the request of an internet research agency (Rakuten Insight, Inc., Tokyo, Japan).

### Study design, data sources, and participants

This cross-sectional study used a dataset from the Japan COVID-19 and Society Internet Survey (JACSIS) (https://jacsis-study.jp/), which is a web-based self-administered questionnaire survey. The JACSIS conducts a survey annually and targets approximately 2 million panellists registered with an internet research agency (Rakuten Insight, Inc., Tokyo, Japan). In the JACSIS, the target population was individuals aged 15–79 years who lived in Japan at a survey. The survey in this study was launched on 27 September 2021 and concluded on 29 October 2021. Of all previous respondents of the previous wave of the JACSIS, an Internet research agency followed up with 33,081 individuals who were available to respond. From 27 September to 29 October 2021, a follow-up survey was conducted among them, resulting in responses from 22,838 individuals (response rate = 69%). Additionally, to achieve a total of 31,000 participants, an additional survey using the same questionnaire was conducted from 23 to 28 October 2021. Consequently, an extra 8,162 responses were collected.

Participants who provided fraudulent responses were excluded to ensure the quality of the data. Three exclusion criteria were employed to identify fraudulent responses: an incorrect answer to an attention check question ("Please select the second last option from the following choices."), straight-line responses regarding drug use and comorbidities, leading to the exclusion of 2,825 respondents. Thus, the Analytic population was 28,175.

### Dependent variable: The number of COVID-19 conspiracy beliefs

To assess conspiracy beliefs about COVID-19, we used three questions from the Oxford Coronavirus Explanations, Attitudes, and Narratives Survey (OCEANS) [27]. The questions were as follows: big Pharma created COVID-19 to profit from the vaccines; COVID-19 was created to force everyone to get vaccinated; the vaccine will be used to carry out mass sterilisation. The available response options were "1. Strongly agree", "2. Somewhat agree ", "3. Neither agree nor disagree", "4. Somewhat disagree", and "5. Strongly disagree". We defined the first and second

options as believing in conspiracy theories about COVID-19 vaccines. The Cronbach's alpha was 0.889 and the Pearson's correlation matrix of the measure is shown in S1 Table.

## Independent variables

Based on our conceptual framework, we selected variables potentially associated with COVID-19 conspiracy beliefs: general vaccine conspiracy beliefs, sociodemographic variables, information sources for COVID-19, trust in authorities, fear of COVID-19, and others.

**1. General vaccine conspiracy beliefs.** We calculated the number of conspiracy beliefs about general vaccines using a Vaccine Conspiracy Beliefs Scale (VCBS) involving 7 items [28]. The questions of VCBS were below: vaccine safety data is often fabricated; immunizing children is harmful and this fact is covered up; pharmaceutical companies cover up the dangers of vaccines; people are deceived about vaccine efficacy; vaccine efficacy data is often fabricated; people are deceived about vaccine safety; the government is trying to cover up the link between vaccines and autism. Potential responses were a 7-point scale that ranged from "1. strongly disagree" to "7. strongly agree". We defined the fifth to seventh options as believing conspiracy theories about general vaccines. The Cronbach's alpha was 0.947 and the Pearson's correlation matrix of the measure is shown in S2 Table.

**2. Sociodemographic variables.** We included age (16–19, 20–24, 25–29, 30–34, 35–39, 40–44, 45–49, 50–54, 55–59, 60–64, 65–69, 70–74, and 75–81 years), sex (men and women), marital status (married, unmarried, widowed, and divorced), educational attainment (lower secondary school, upper secondary school, specialised training college [post-secondary courses], junior college and college of technology, university, and master's or doctor's degree), employment status (regular employee, temporary employee, self-employed, employer, student, unemployed or retired, home maker, and others), annual household income (0 to <2, 2 to <4, 4 to <5, 5 to <8, and ≥8 million yen), household financial assets (0 to <1, 1 to <4, 4 to <9, 9 to <20, and ≥20 million yen), and household indebtedness (none, >0 to <2, and ≥2 million yen).

**3. Information sources for COVID-19.** As for sources of information about COVID-19, we targeted the following 14 information sources: websites of government agencies, websites of research institutions, video sharing platforms (e.g. YouTube), LINE, Twitter, Facebook, Instagram, web news, newspapers, magazines, books, TV news, and tabloid TV shows. Sources of information for COVID-19 were assessed by the following question: "Have you obtained medical information about health topics, including COVID-19, from each of the following sources?" The options available for each source were "Yes" and "No". If respondents chose "Yes", they were asked that "How much do you trust the source of information you selected in the previous question?" The possible options were "1. Very trust", " 2. Trust", "3. Somewhat trust", "4. Somewhat distrust", "5. Distrust", and " 6. Distrust at all". We defined the first to third options as "trusted", and the fourth to sixth as "distrusted". Combining the former and latter questions, we defined three categories: "Not use", "Use but distrust", and "Use and trust".

**4. Trust in authorities.** To measure the level of trust in authorities, we targeted the following 3 authorities: the government of Japan, prefectural administrations, and municipal administrations. Trust in authorities was measured using the following questions: the government is trustworthy; the administration of the municipality in which you currently live is trustworthy; the administration of the prefecture in which you currently live is trustworthy. The response options were "1. Agree", "2. Somewhat agree", "3. Somewhat disagree", and "4. Disagree". We defined the first and second options as "Trust", and the third and fourth as "Distrust".

**5. Fear of COVID-19.** We used the Japanese Version of the Fear of Coronavirus Disease 2019 Scale (FCV-19S) [29]. This scale involves 7 items with a scale of 1 (strongly disagree) to 5

(strongly agree) points. The range of the scale was from 7 to 35 points. We adopted ≥21 of a cut-off point to identify psychological distress or difficulties in daily living due to the fear of COVID-19 [30].

**6. Other covariates.** Experiencing discrimination regarding COVID-19 was obtained using a question: "In the last two months, have you experienced any of the following events? Felt discrimination related to COVID-19." Available options were "1. Yes (experienced for the first time in the last two months)", "2. Yes (and had happened before)", "3. Never (but had happened before)", and "4. Never happened before". We defined the first to third options as "Experienced", and the last option as "Never". Additionally, we obtained a medical history of depression, other mental disorders, and coronavirus infections.

## Statistical analysis

We employed linear regression analysis with a robust error variance to estimate the unstandardised coefficients (Bs) of the number of conspiracy beliefs regarding COVID-19. Although the distribution of a dependent variable is right-skewed, linear regression models can provide valid estimations [31]. A positive coefficient means holding more conspiracy theories while a negative coefficient means holding fewer conspiracy theories. We created a fully adjusted model using the simultaneous forced-entry method. The fully adjusted model included all independent variables simultaneously.

To impute missing values, we used a k-nearest neighbour imputation (the R package "VIM"). After the imputation, sampling weights were calculated using the raking method from the R package "anesrake" to match the proportions of age, sex, prefecture of residence, marital status, annual household income, and educational attainment from census data. Detailed information on the sampling weights has been described in S3 Table.

A P-value <0.05 (2-tailed) was considered statistically significant. All analyses were conducted in R (ver. 4.3.0; R Foundation for Statistical Computing, Vienna, Austria).

## Results

Table 1 shows basic participant characteristics (full table in S4 Table). The median age of respondents was 50, ranging from 16 to 81. The proportion of women was 50.8%. After imputation and weighting, 10.6% (95% confidence interval [CI] 10.1, 11.1) of the participants believed in one COVID-19 conspiracy theory (OCEANS), 6.5% (95%CI 6.1, 6.9) in two, and 7.3% (95%CI 7.0, 7.8) in three. Thus, the prevalence of harboring at least one COVID-19 conspiracy theory was 24.4% (95%CI 23.7, 25.1). The detailed results of the original categories for three questions on COVID-19 conspiracy beliefs are shown in S5 Table. On the other hand, participants who believed in at least one conspiracy theory regarding general vaccines (VCBS) constituted 33.6% (detailed results are shown in S6 Table).

Table 2 presents the weighted mean of the number of COVID-19 conspiracy beliefs (full table in S7 Table). University graduates exhibited a higher endorsement of COVID-19 conspiracy beliefs (0.49), and the highest annual household income group also showed higher conspiracy beliefs (0.52). Paradoxically, participants who distrusted the Japanese government endorsed fewer conspiracy beliefs (0.40) compared to those who trusted the government (0.56).

Table 3 shows the results of multivariable linear regression analysis with sampling weights after imputation (full table in S8 Table). Lower secondary education as well as holding a master's or doctor's degree were associated with endorsing fewer COVID-19 conspiracy theories compared with upper secondary education (lower secondary education: B -0.089, 95%CI -0.172, -0.006; master's or doctor's degree: B -0.066, 95%CI -0.126, -0.006). Temporary

**Table 1. Basic characteristics of 28,175 participants.**

| Variable | Response participants | | Imputed population | | Weighted imputed population | |
|---|---|---|---|---|---|---|
| | n = 28,175 | (%) | n = 28,175 | (%) | n = 28,175 | (%) |
| The number of conspiracy theory beliefs regarding COVID-19 | | | | | | |
| 0 | 21,414 | (76.0%) | 21,414 | (76.0%) | 21,297 | (75.6%) |
| 1 | 2,876 | (10.2%) | 2,876 | (10.2%) | 2,974 | (10.6%) |
| 2 | 1,830 | (6.5%) | 1,830 | (6.5%) | 1,836 | (6.5%) |
| 3 | 2,055 | (7.3%) | 2,055 | (7.3%) | 2,068 | (7.3%) |
| The number of conspiracy theory beliefs regarding general vaccines | | | | | | |
| 0 | 18,798 | (66.7%) | 18,798 | (66.7%) | 18,712 | (66.4%) |
| 1 | 3,021 | (10.7%) | 3,021 | (10.7%) | 3,010 | (10.7%) |
| 2 | 2,022 | (7.2%) | 2,022 | (7.2%) | 2,067 | (7.3%) |
| 3 | 1,180 | (4.2%) | 1,180 | (4.2%) | 1,203 | (4.3%) |
| 4 | 873 | (3.1%) | 873 | (3.1%) | 865 | (3.1%) |
| 5 | 772 | (2.7%) | 772 | (2.7%) | 768 | (2.7%) |
| 6 | 741 | (2.6%) | 741 | (2.6%) | 779 | (2.8%) |
| 7 | 768 | (2.7%) | 768 | (2.7%) | 770 | (2.7%) |
| Age (years old) | | | | | | |
| 16–19 | 573 | (2.0%) | 573 | (2.0%) | 949 | (3.4%) |
| 20–24 | 1,829 | (6.5%) | 1,829 | (6.5%) | 1,801 | (6.4%) |
| 25–29 | 1,797 | (6.4%) | 1,797 | (6.4%) | 1,834 | (6.5%) |
| 30–34 | 1,963 | (7.0%) | 1,963 | (7.0%) | 1,885 | (6.7%) |
| 35–39 | 2,186 | (7.8%) | 2,186 | (7.8%) | 2,114 | (7.5%) |
| 40–44 | 2,600 | (9.2%) | 2,600 | (9.2%) | 2,350 | (8.3%) |
| 45–49 | 2,854 | (10.1%) | 2,854 | (10.1%) | 2,798 | (9.9%) |
| 50–54 | 2,529 | (9.0%) | 2,529 | (9.0%) | 2,661 | (9.4%) |
| 55–59 | 2,256 | (8.0%) | 2,256 | (8.0%) | 2,250 | (8.0%) |
| 60–64 | 2,234 | (7.9%) | 2,234 | (7.9%) | 2,125 | (7.5%) |
| 65–69 | 2,648 | (9.4%) | 2,648 | (9.4%) | 2,263 | (8.0%) |
| 70–74 | 2,655 | (9.4%) | 2,655 | (9.4%) | 2,781 | (9.9%) |
| 75–81 | 2,051 | (7.3%) | 2,051 | (7.3%) | 2,362 | (8.4%) |
| Sex | | | | | | |
| Men | 13,870 | (49.2%) | 13,870 | (49.2%) | 13,986 | (49.6%) |
| Women | 14,305 | (50.8%) | 14,305 | (50.8%) | 14,189 | (50.4%) |
| Marital status | | | | | | |
| Married | 16,969 | (60.2%) | 16,969 | (60.2%) | 17,472 | (62.0%) |
| Unmarried | 8,322 | (29.5%) | 8,322 | (29.5%) | 7,918 | (28.1%) |
| Widowed | 1,001 | (3.6%) | 1,001 | (3.6%) | 1,200 | (4.3%) |
| Divorced | 1,883 | (6.7%) | 1,883 | (6.7%) | 1,585 | (5.6%) |
| Educational attainment | | | | | | |
| Lower Secondary School | 392 | (1.4%) | 392 | (1.4%) | 1,960 | (7.0%) |
| Upper Secondary School | 7,592 | (27.1%) | 7,638 | (27.1%) | 12,780 | (45.4%) |
| Specialised Training College (Post-Secondary Courses) | 3,203 | (11.4%) | 3,230 | (11.5%) | 2,775 | (9.9%) |
| Junior College and College of Technology | 3,045 | (10.9%) | 3,065 | (10.9%) | 2,447 | (8.7%) |
| University | 12,368 | (44.2%) | 12,440 | (44.2%) | 7,421 | (26.3%) |
| Master's or doctor's degree | 1,409 | (5.0%) | 1,410 | (5.0%) | 791 | (2.8%) |
| (Missing) | 166 | | | | | |
| Employment status | | | | | | |
| Regular employee | 9,722 | (34.5%) | 9,722 | (34.5%) | 9,231 | (32.8%) |

(*Continued*)

**Table 1.** (Continued)

| Variable | Response participants | | Imputed population | | Weighted imputed population | |
|---|---|---|---|---|---|---|
| | n = 28,175 | (%) | n = 28,175 | (%) | n = 28,175 | (%) |
| *Temporary employee* | 4,881 | (17.3%) | 4,881 | (17.3%) | 4,976 | (17.7%) |
| *Self-employed* | 1,732 | (6.1%) | 1,732 | (6.1%) | 1,659 | (5.9%) |
| *Employer* | 990 | (3.5%) | 990 | (3.5%) | 980 | (3.5%) |
| *Student* | 1,267 | (4.5%) | 1,267 | (4.5%) | 1,387 | (4.9%) |
| *Unemployed or retired* | 4,636 | (16.5%) | 4,636 | (16.5%) | 4,768 | (16.9%) |
| *Home maker* | 4,711 | (16.7%) | 4,711 | (16.7%) | 4,938 | (17.5%) |
| *Others* | 236 | (0.8%) | 236 | (0.8%) | 235 | (0.8%) |
| Annual household income | | | | | | |
| *0 to <2 million yen* | 2,415 | (10.9%) | 2,575 | (9.1%) | 3,298 | (11.7%) |
| *2 to <4 million yen* | 5,738 | (25.9%) | 7,142 | (25.3%) | 8,298 | (29.5%) |
| *4 to <5 million yen* | 2,868 | (13.0%) | 3,908 | (13.9%) | 3,669 | (13.0%) |
| *5 to <8 million yen* | 6,082 | (27.5%) | 8,530 | (30.3%) | 7,219 | (25.6%) |
| *≥8 million yen* | 5,024 | (22.7%) | 6,020 | (21.4%) | 5,692 | (20.2%) |
| *(Missing)* | 6,048 | | | | | |
| Household financial assets | | | | | | |
| *0 to <1 million yen* | 3,034 | (17.5%) | 4,330 | (15.4%) | 5,144 | (18.3%) |
| *1 to <4 million yen* | 3,779 | (21.8%) | 6,707 | (23.8%) | 7,434 | (26.4%) |
| *4 to <9 million yen* | 3,440 | (19.9%) | 6,594 | (23.4%) | 6,245 | (22.2%) |
| *9 to <20 million yen* | 3,247 | (18.7%) | 5,349 | (19.0%) | 4,931 | (17.5%) |
| *≥20 million yen* | 3,825 | (22.1%) | 5,195 | (18.4%) | 4,422 | (15.7%) |
| *(Missing)* | 10,850 | | | | | |
| Household indebtedness | | | | | | |
| *None* | 15,722 | (71.5%) | 20,145 | (71.5%) | 19,914 | (70.7%) |
| *>0 to <2 million yen* | 1,510 | (6.9%) | 2,005 | (7.1%) | 2,236 | (7.9%) |
| *≥2 million yen* | 4,771 | (21.7%) | 6,025 | (21.4%) | 6,025 | (21.4%) |
| *(Missing)* | 6,172 | | | | | |

Full table is in S4 Table.

employees (B -0.051, 95%CI -0.095, -0.007), self-employed (B -0.102, 95%CI -0.162, -0.042), unemployed or retired (B -0.078, 95%CI -0.131, -0.025), and home makers (B -0.086, 95%CI -0.135, -0.037) held fewer conspiracy theories compared to regular employees. In addition, higher household income and assets were also associated with endorsing more conspiracy theories (annual household income of ≥8 million yen: B 0.069, 95%CI 0.020, 0.117 vs. 4 to <5 million yen; household financial assets of 9 to <20 million yen: B 0.060, 95%CI 0.010, 0.109 vs. 0 to <1 million yen).

As for information sources, participants who used and trusted the websites of government agencies had fewer conspiracy beliefs than those who did not use the websites of government (B -0.109, 95%CI -0.139, -0.079). On the other hand, participants who used and trusted video sharing platforms (e.g. YouTube) (B 0.089, 95%CI 0.038, 0.139), LINE (B 0.053, 95%CI 0.009, 0.097), Facebook (B 0.089, 95%CI 0.000, 0.178), and Instagram (B 0.093, 95%CI 0.008, 0.178) endorsed more conspiracy theories than those who did not use them. Participants who trusted the information from books endorsed more conspiracy beliefs than those who did not use books (B 0.136, 95%CI 0.068, 0.204). Participants who trusted TV news endorsed fewer

**Table 2. Weighted means of the number of COVID-19 conspiracy beliefs after imputation.**

| Variable | Mean number of conspiracy theory beliefs regarding COVID 19 |
|---|---|
| The number of conspiracy theory beliefs regarding general vaccines | |
| 0 | 0.41 |
| 1 | 0.48 |
| 2 | 0.45 |
| 3 | 0.45 |
| 4 | 0.56 |
| 5 | 0.50 |
| 6 | 0.75 |
| 7 | 1.12 |
| Educational attainment | |
| Lower Secondary School | 0.39 |
| Upper Secondary School | 0.45 |
| Specialised Training College (Post-Secondary Courses) | 0.46 |
| Junior College and College of Technology | 0.44 |
| University | 0.49 |
| Master's or doctor's degree | 0.42 |
| Employment status | |
| Regular employee | 0.51 |
| Temporary employee | 0.41 |
| Self-employed | 0.44 |
| Employer | 0.58 |
| Student | 0.52 |
| Unemployed or retired | 0.45 |
| Home maker | 0.38 |
| Others | 0.38 |
| Annual household income | |
| 0 to <2 million yen | 0.44 |
| 2 to <4 million yen | 0.45 |
| 4 to <5 million yen | 0.42 |
| 5 to <8 million yen | 0.44 |
| ≥8 million yen | 0.52 |
| Household financial assets | |
| 0 to <1 million yen | 0.41 |
| 1 to <4 million yen | 0.48 |
| 4 to <9 million yen | 0.46 |
| 9 to <20 million yen | 0.50 |
| ≥20 million yen | 0.42 |
| Household indebtedness | |
| None | 0.45 |
| >0 to <2 million yen | 0.47 |
| ≥2 million yen | 0.48 |
| Information source for COVID-19: Websites of government agencies | |
| Not use | 0.48 |
| Use but distrust | 0.66 |
| Use and trust | 0.41 |
| Information source for COVID-19: Video sharing platforms (e.g. YouTube) | |

*(Continued)*

**Table 2.** (Continued)

| Variable | Mean number of conspiracy theory beliefs regarding COVID 19 |
|---|---|
| *Not use* | 0.43 |
| *Use but distrust* | 0.50 |
| *Use and trust* | 0.65 |
| Information source for COVID-19: LINE | |
| *Not use* | 0.43 |
| *Use but distrust* | 0.57 |
| *Use and trust* | 0.56 |
| Information source for COVID-19: Twitter | |
| *Not use* | 0.44 |
| *Use but distrust* | 0.48 |
| *Use and trust* | 0.60 |
| Information source for COVID-19: Facebook | |
| *Not use* | 0.44 |
| *Use but distrust* | 0.66 |
| *Use and trust* | 0.77 |
| Information source for COVID-19: Instagram | |
| *Not use* | 0.44 |
| *Use but distrust* | 0.60 |
| *Use and trust* | 0.74 |
| Information source for COVID-19: Books | |
| *Not use* | 0.43 |
| *Use but distrust* | 0.82 |
| *Use and trust* | 0.72 |
| Information source for COVID-19: TV news | |
| *Not use* | 0.54 |
| *Use but distrust* | 0.48 |
| *Use and trust* | 0.43 |
| Information source for COVID-19: Tabloid TV shows | |
| *Not use* | 0.46 |
| *Use but distrust* | 0.43 |
| *Use and trust* | 0.46 |
| Trust in the government of Japan | |
| *Distrust* | 0.40 |
| *Trust* | 0.56 |

Full table is in S7 Table.

conspiracy theories (B -0.081, 95%CI -0.128, -0.035 vs. not used), while those who trusted tabloid TV shows held more conspiracy theories (B 0.051, 95%CI 0.015, 0.087 vs. not used).

Trust in authorities was associated with endorsing COVID-19 conspiracy beliefs. Trust in the government of Japan was associated with holding more conspiracy theories (B 0.175, 95% CI 0.139, 0.211), compared to those who distrusted government.

## Discussion

From a survey involving 28,175 residents of Japan, we estimated that 24.4% of individuals in Japan endorsed at least one COVID-19 conspiracy theory. The prevalence of COVID-19

**Table 3. Associations between independent variables and the number of COVID-19 conspiracy beliefs.**

| Variable | Weighted multivariable adjusted model (n = 28,175) | |
| --- | --- | --- |
| | Beta | 95% confidence interval |
| The number of conspiracy theory beliefs regarding general vaccines | | |
| *0 (reference)* | — | — |
| *1* | 0.059 | 0.017, 0.102 |
| *2* | 0.045 | -0.004, 0.094 |
| *3* | 0.040 | -0.022, 0.103 |
| *4* | 0.178 | 0.098, 0.258 |
| *5* | 0.128 | 0.042, 0.215 |
| *6* | 0.401 | 0.299, 0.503 |
| *7* | 0.718 | 0.593, 0.843 |
| Educational attainment | | |
| *Upper Secondary School (reference)* | — | — |
| *Lower Secondary School* | -0.089 | -0.172, -0.006 |
| *Specialised Training College (Post-Secondary Courses)* | 0.018 | -0.026, 0.061 |
| *Junior College and College of Technology* | 0.034 | -0.008, 0.077 |
| *University* | 0.009 | -0.024, 0.042 |
| *Master's or doctor's degrees* | -0.066 | -0.126, -0.006 |
| Employment status | | |
| *Regular employee (reference)* | — | — |
| *Temporary employee* | -0.051 | -0.095, -0.007 |
| *Self-employed* | -0.102 | -0.162, -0.042 |
| *Employer* | 0.004 | -0.081, 0.089 |
| *Student* | -0.061 | -0.155, 0.034 |
| *Unemployed or retired* | -0.078 | -0.131, -0.025 |
| *Home maker* | -0.086 | -0.135, -0.037 |
| *Others* | -0.118 | -0.248, 0.012 |
| Annual household income | | |
| *4 to <5 million yen (reference)* | — | — |
| *0 to <2 million yen* | 0.038 | -0.024, 0.101 |
| *2 to <4 million yen* | 0.035 | -0.008, 0.078 |
| *5 to <8 million yen* | 0.016 | -0.026, 0.058 |
| *≥8 million yen* | 0.069 | 0.020, 0.117 |
| Household financial assets | | |
| *0 to <1 million yen (reference)* | — | — |
| *1 to <4 million yen* | 0.039 | -0.006, 0.084 |
| *4 to <9 million yen* | 0.020 | -0.026, 0.066 |
| *9 to <20 million yen* | 0.060 | 0.010, 0.109 |
| *≥20 million yen* | -0.016 | -0.068, 0.036 |
| Household indebtedness | | |
| *None (reference)* | — | — |
| *>0 to <2 million yen* | -0.025 | -0.080, 0.030 |
| *≥2 million yen* | -0.016 | -0.052, 0.020 |
| Information source for COVID-19: Websites of government agencies | | |
| *Not used (reference)* | — | — |
| *Use but distrust* | 0.083 | -0.011, 0.178 |
| *Use and trust* | -0.109 | -0.139, -0.079 |

(*Continued*)

**Table 3.** (Continued)

| Variable | Weighted multivariable adjusted model | |
|---|---|---|
| | **(n = 28,175)** | |
| | **Beta** | **95% confidence interval** |
| Information source for COVID-19: Video sharing platforms (e.g. YouTube) | | |
| *Not used (reference)* | — | — |
| *Use but distrust* | -0.002 | -0.073, 0.070 |
| *Use and trust* | 0.089 | 0.038, 0.139 |
| Information source for COVID-19: LINE | | |
| *Not used (reference)* | — | — |
| *Use but distrust* | 0.064 | -0.029, 0.157 |
| *Use and trust* | 0.053 | 0.009, 0.097 |
| Information source for COVID-19: Twitter | | |
| *Not used (reference)* | — | — |
| *Use but distrust* | -0.002 | -0.069, 0.066 |
| *Use and trust* | 0.048 | -0.005, 0.102 |
| Information source for COVID-19: Facebook | | |
| *Not used (reference)* | — | — |
| *Use but distrust* | 0.042 | -0.079, 0.163 |
| *Use and trust* | 0.089 | 0.000, 0.178 |
| Information source for COVID-19: Instagram | | |
| *Not used (reference)* | — | — |
| *Use but distrust* | -0.012 | -0.120, 0.095 |
| *Use and trust* | 0.093 | 0.008, 0.178 |
| Information source for COVID-19: Books | | |
| *Not used (reference)* | — | — |
| *Use but distrust* | 0.137 | -0.011, 0.286 |
| *Use and trust* | 0.136 | 0.068, 0.204 |
| Information source for COVID-19: TV news | | |
| *Not used (reference)* | — | — |
| *Use but distrust* | -0.040 | -0.102, 0.022 |
| *Use and trust* | -0.081 | -0.128, -0.035 |
| Information source for COVID-19: Tabloid TV shows | | |
| *Not used (reference)* | — | — |
| *Use but distrust* | 0.005 | -0.043, 0.053 |
| *Use and trust* | 0.051 | 0.015, 0.087 |
| Trust in the government of Japan | | |
| *Distrust (reference)* | — | — |
| *Trust* | 0.175 | 0.139, 0.211 |

Full table is in S8 Table.

Linear regression analysis with a robust error variance was used, applying sampling weights and imputation.

A positive coefficient means holding more conspiracy theories while a negative coefficient means holding fewer conspiracy theories.

The adjusted model included all independent variables simultaneously.

conspiracy beliefs worldwide ranges from 5.0% to 39.0% [8, 12]. Surprisingly, we found that the characteristics of respondents endorsing conspiracy beliefs were often the opposite of those reported in studies from Western contexts. For example, people from higher socioeconomic backgrounds–such as higher income, higher wealth, and regular employment–were more likely to endorse COVID-19 conspiracy beliefs. Furthermore, 62.7% of respondents distrusted the government of Japan but they tended to hold fewer COVID-19 conspiracy theories.

## Interpretations of the results

In our study, individuals who trusted the government were more likely to believe in conspiracy theories. This finding is in contrast to previous research in other countries [12, 13] and is therefore unique. A prior study from JACSIS suggests that individuals with high trust in the government of Japan tended to practice preventive measures more often than those with low trust [32]. Therefore, we expected that individuals with high trust in the government would also disbelieve conspiracy theories. However, our results suggest the opposite–individuals sceptical of the government might have been more rational and critical about conspiracy theories, but not necessarily about preventive measures. In addition, we found that holding more COVID-19 conspiracy beliefs was associated with LOWER vaccine hesitancy (S9 Table). However, if respondents simultaneously endorsed more general vaccine and COVID-19 conspiracy beliefs, holding more COVID-19 conspiracy beliefs was associated with vaccine hesitancy (S10 Table). These findings suggest that different types of conspiracy theories, such as those related to general vaccines and COVID-19, may interact in complex ways to shape health-related behaviour. Moreover, a previous study in Japan indicates that belief in one COVID-19 conspiracy theory was associated with booster vaccine hesitancy in 2022 [33]. The role of COVID-19 conspiracy theories in Japan may have changed over time. Future studies are needed to investigate the long-term effects of COVID-19 conspiracy theories on vaccine hesitancy and health-related behaviour.

Prior studies also suggest that individuals with high socioeconomic status are more likely to disbelieve conspiracy theories [12, 13]. Indeed, our results suggest that the highest educational attainment (a master's or doctor's degree) was associated with holding fewer COVID-19 conspiracy theories compared with the middle educational attainment (upper secondary school, which is the most standard educational attainment in Japan). However, individuals with the lowest educational attainment (lower secondary school) also endorsed fewer conspiracy theories than those with middle educational attainment. In addition, higher socioeconomic position–based on higher income, higher wealth, and regular employment–were associated with endorsing more COVID-19 conspiracy theories. This may reflect the nature of Japanese society. In Japanese society, it has been previously noted that conspiracy beliefs often attract people from higher socioeconomic backgrounds [34]. For example, the religious cult, Aum Shinrikyo (responsible for the 1995 Tokyo subway sarin attack) promulgated antisemitic conspiracy theories, claiming that the government of Japan was secretly manipulated by Jews and Freemasons [35]. Members of the cult who planned and participated in the sarin gas attack were drawn from the elite including those with expertise in medicine, biology, chemistry, physics, and engineering [36]. Future research must identify the mechanisms behind this unique Japanese result.

The results regarding information sources in this study, as well as those of previous research [12, 13, 18], indicated that individuals trusting social media (Twitter, Facebook, and Instagram) were more likely to believe in conspiracy theories while those who believed in mainstream sources such as government and traditional media news tended to endorse fewer conspiracy theories. This finding may be attributed to the proliferation of conspiracy theories on social media in Japan [37]. However, it's important to note that the causal relationship between social media use and conspiracy beliefs is still controversial [7, 18]. Regarding TV

programs in Japan, trusting TV news was associated with disbelief in conspiracy theories. Although TV news has provided biased reports on vaccines; however, in the COVID-19 case, they seem to reflect the opinions of the government, authorities, and expert groups [38]. On the other hand, tabloid TV shows featured celebrities, critics, journalists, and experts who were unfamiliar with public health [39]. Therefore, the observed difference between TV news and tabloid TV shows could be due to these features. Believing in information from books was associated with endorsing COVID-19 conspiracy theories. Misinformation, especially about healthcare, thrives on books in Japan [40–42]. Because these books were bestsellers, some people might have purchased and believed them. On the other hand, individuals who believe in conspiracy theories may have actively purchased books supporting their beliefs. The correlation between books and endorsing conspiracy theories might be attributed to the proliferation of books containing misinformation.

## Strengths and limitations

The limitations of this study should be noted. First, this study was cross-sectional, which means that a temporal relationship between the independent variables and the outcome remains unclear. Second, our examination of COVID-19 conspiracy theories relied on three questions. The estimated prevalence of COVID-19 conspiracy beliefs may be underestimated. However, the estimated prevalence was relatively similar to results from previous systematic reviews [8, 12]. This study also has strengths. Unlike many previous studies with sample sizes over 200 [13], our study included 28,175 participants. Although recruitment through an internet research agency can limit the generalizability of the findings to the Japanese population, the data include a diverse sample of residents of Japan, including a substantial representation of those with low socioeconomic status. In addition, we employed sampling weights based on census data to improve external validity. Moreover, this study explored numerous potential correlates, enabling a comprehensive assessment of the characteristics of individuals in Japan who endorse COVID-19 conspiracy theories. These strengths can enhance the generalizability of the findings.

## Conclusions

Conspiracy theories are a global problem and pose a public health risk. This study reported the prevalence of COVID-19 conspiracy beliefs and their correlates in Japan. Some of the results were different, while some replicated findings from Western countries. Similar to findings from previous research, individuals who used social media as a source of information endorsed conspiracy theories. However, contrary to the results from Western countries, individuals with the lowest level of education were less likely to believe in conspiracy theories. Furthermore, higher socioeconomic residents of Japan–higher income, higher wealth, and regular employment–are more likely to endorse conspiracy beliefs about COVID-19. Those who trusted the government were more likely to endorse conspiracy beliefs. Our findings might support that combating the spread of conspiracy theories requires interventions that take into account cultural contexts. Furthermore, a key area for future research is to examine the mechanisms underpinning these observations specific to Japan. This knowledge gap should be addressed to understand the phenomenon.

## Supporting information

**S1 Table. Pearson's correlation matrix of three COVID-19 conspiracy belief questions from the Oxford Coronavirus Explanations, Attitudes, and Narratives Survey (OCEANS).** (PDF)

**S2 Table. Pearson's correlation matrix of seven general vaccine conspiracy belief questions from the Vaccine Conspiracy Beliefs Scale (VCBS).**
(PDF)

**S3 Table. Descriptive statistics of the results with sampling weights.**
(PDF)

**S4 Table. Full table of basic participant characteristics.**
(PDF)

**S5 Table. Descriptive statistics of the original categories for three questions on COVID-19 conspiracy beliefs from the Oxford Coronavirus Explanations, Attitudes, and Narratives Survey (OCEANS) after applying sampling weights.**
(PDF)

**S6 Table. Descriptive statistics of the original categories for seven questions on general vaccine conspiracy beliefs from the Vaccine Conspiracy Beliefs Scale (VCBS) after applying sampling weights.**
(PDF)

**S7 Table. Full table of the weighted mean of the number of COVID-19 conspiracy beliefs.**
(PDF)

**S8 Table. Full table of associations between independent variables and the number of COVID-19 conspiracy beliefs.**
(PDF)

**S9 Table. Associations between independent variables and COVID-19 vaccine hesitancy (0 = intend, 1 = hesitant) from weighted Poisson regression analyses with a robust error variance after imputation.**
(PDF)

**S10 Table. Associations of the interaction term of conspiracy beliefs about COVI-19 and conspiracy beliefs about general vaccines with COVID-19 vaccine hesitancy (0 = intend, 1 = hesitant) from a weighted Poisson regression analysis with a robust error variance after imputation.**
(PDF)

## Acknowledgments

We thank all the participants who voluntarily shared their time and experience for the JACSIS.

## Author Contributions

**Conceptualization:** Yukihiro Sato, Ichiro Kawachi, Yasuaki Saijo, Eiji Yoshioka, Ken Osaka, Takahiro Tabuchi.

**Data curation:** Yukihiro Sato, Takahiro Tabuchi.

**Formal analysis:** Yukihiro Sato, Ichiro Kawachi, Yasuaki Saijo, Eiji Yoshioka, Ken Osaka, Takahiro Tabuchi.

**Funding acquisition:** Takahiro Tabuchi.

**Investigation:** Takahiro Tabuchi.

**Methodology:** Yukihiro Sato, Ichiro Kawachi, Yasuaki Saijo, Eiji Yoshioka, Ken Osaka, Takahiro Tabuchi.

**Project administration:** Takahiro Tabuchi.

**Resources:** Takahiro Tabuchi.

**Software:** Yukihiro Sato.

**Supervision:** Ichiro Kawachi, Yasuaki Saijo, Eiji Yoshioka, Ken Osaka, Takahiro Tabuchi.

**Validation:** Ichiro Kawachi, Yasuaki Saijo, Eiji Yoshioka, Ken Osaka, Takahiro Tabuchi.

**Visualization:** Yukihiro Sato.

**Writing – original draft:** Yukihiro Sato.

**Writing – review & editing:** Ichiro Kawachi, Yasuaki Saijo, Eiji Yoshioka, Ken Osaka, Takahiro Tabuchi.

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
