## [Decision Letter · Decision Letter 0]

20 Oct 2024

PONE-D-24-38558

Correlates of COVID-19 conspiracy theory beliefs in Japan: A nationally-representative cross-sectional survey

PLOS ONE

Dear Dr. Sato,

Thank you for submitting your manuscript to PLOS ONE. After careful consideration, we feel that it has merit but does not fully meet PLOS ONE’s publication criteria as it currently stands. Therefore, we invite you to submit a revised version of the manuscript that addresses the points raised during the review process.

We look forward to receiving your revised manuscript.

Kind regards,

Osmond Ekwebelem

Academic Editor

PLOS ONE

2. In the online submission form you indicate that your data is not available for proprietary reasons and have provided a contact point for accessing this data. Please note that your current contact point is a co-author on this manuscript. According to our Data Policy, the contact point must not be an author on the manuscript and must be an institutional contact, ideally not an individual. Please revise your data statement to a non-author institutional point of contact, such as a data access or ethics committee, and send this to us via return email. Please also include contact information for the third party organization, and please include the full citation of where the data can be found.

3. We note that there is identifying data in the Supporting Information file < 2_Suppl.pdf>. Due to the inclusion of these potentially identifying data, we have removed this file from your file inventory. Prior to sharing human research participant data, authors should consult with an ethics committee to ensure data are shared in accordance with participant consent and all applicable local laws.

-Location data

Please remove or anonymize all personal information (Ages more specific than whole numbers), ensure that the data shared are in accordance with participant consent, and re-upload a fully anonymized data set. Please note that spreadsheet columns with personal information must be removed and not hidden as all hidden columns will appear in the published file.

Reviewers' comments:

Reviewer's Responses to Questions

**Comments to the Author**

1. Is the manuscript technically sound, and do the data support the conclusions?

Reviewer #1: No

Reviewer #2: Yes

Reviewer #3: Partly

Reviewer #4: Yes

2. Has the statistical analysis been performed appropriately and rigorously? 

Reviewer #1: N/A

Reviewer #2: Yes

Reviewer #3: Yes

Reviewer #4: Yes

3. Have the authors made all data underlying the findings in their manuscript fully available?

Reviewer #1: No

Reviewer #2: Yes

Reviewer #3: No

Reviewer #4: No

4. Is the manuscript presented in an intelligible fashion and written in standard English?

Reviewer #1: No

Reviewer #2: Yes

Reviewer #3: Yes

Reviewer #4: Yes

5. Review Comments to the Author

Reviewer #1: I understand they have many results, but understanding can be challenging. This article should be shorter and easier to understand. I have the following comments:

1- Language: Better language (letter errors, grammar... etc.) is needed. It must use the Grammarly program to become a better manuscript.

2. Methodology: There are a lot of references. It must reduce references in the methodology

2- The result is a lot of mixing and results found in this manuscript. Most of the data is old, as shown: There are very long tables, and many tables are also in the appendix. It isn't easy to understand these tables.

Discussion: There is no new result to discuss. Most of these results have been published previously. I can't see new results, and this data is old in use.

Therefore, based on the issues identified, the manuscript is not currently suitable for publication in this journal. Significant revisions are necessary to address the concerns raised.

It's crucial to condense the results (tables) and enhance the explanation in the discussion and methodology. These improvements are necessary for the manuscript to be considered for publication.

Reviewer #2: Dear Authors and Research Team,

Your team conducted a social science study on the formation and spread of rumors during the COVID-19 pandemic. This research definitely has important social application value, and I believe it should be published.

However, I still have some questions and suggestions for further consideration and revision.

Major Opinions

1. In the methods section, the "Study Design, Data Sources, and Participants" discusses the survey questionnaire design. Although you have excluded participants who provided fraudulent responses, I would like to know how your questionnaire distinguishes and confirms the nationality of respondents.

2. This survey was launched on September 27, 2021, and concluded on October 29, 2021. You discuss trust in authorities; however, until October 4, 2021, the authorities were part of the Yoshihide Suga Cabinet, and after that, they belonged to the first Kishida Fumio Cabinet. When designing the survey, did you specify which authorities the participants should consider? If not, I think a more detailed analysis is necessary, as this is very important for the credibility of certain perspectives in the results and discussion.

Minor Opinions

1. On page 15, the statement "A broader range of conspiracy theories needs to be considered, including conspiracy theories about vaccines" is ambiguous, making it difficult to understand the authors' specific position and the measures they believe should be taken. The authors should improve the clarity of this expression.

2. In Table 3, the values and headers labeled "B" and "95 CI" are not aligned.

These are my suggestions.

Good luck with your revisions!

Reviewer #3: The introduction sounds quite fragmented. A more cohesive, discursive paragraph is recommended.

Conspiracy believes appeared to have recorded even 42,2%. Add findings from Who Believes in COVID-19 Conspiracy Theories in Croatia? Prevalence and Predictors of Conspiracy Beliefs.

Tonković M, Dumančić F, Jelić M, Čorkalo Biruški D.

Front Psychol. 2021 Jun 18;12:643568. doi: 10.3389/fpsyg.2021.643568. eCollection 2021.

Update in light of the recent publications:

Factors associated with COVID-19 booster vaccine hesitancy: a nationwide, cross-sectional survey in Japan.

Takamatsu A, Honda H, Miwa T, Tabuchi T, Taniguchi K, Shibuya K, Tokuda Y.

Public Health. 2023 Oct;223:72-79. doi: 10.1016/j.puhe.2023.07.022. Epub 2023 Aug 22.

PMID: 37619504

Effects of Health Literacy in the Fight Against the COVID-19 Infodemic: The Case of Japan.

Cheng JW, Nishikawa M.

Health Commun. 2022 Nov;37(12):1520-1533. doi: 10.1080/10410236.2022.2065745. Epub 2022 May 3.

PMID: 35505456

A quantitative content analysis of topical characteristics of the online COVID-19 infodemic in the United States and Japan.

Seah M, Iwakuma M.

BMC Public Health. 2024 Sep 9;24(1):2447. doi: 10.1186/s12889-024-19813-y.

PMID: 39251957 Free PMC article.

Exploratory study to characterise the individual types of health literacy and beliefs and their associations with infection prevention behaviours amid the COVID-19 pandemic in Japan: a longitudinal study.

Yagihashi M, Murakami M, Kato M, Yamamura A, Miura A, Hirai K.

PeerJ. 2024 Feb 22;12:e16905. doi: 10.7717/peerj.16905. eCollection 2024.

PMID: 38406277

Methods

The methodology is well-detailed, and the tools are clearly explained, making the study replicable.

Although the survey involved a significant sample size of 31,000, the article does not specify whether its characteristics were representative of the general population in Japan. Given the correlation with demographic data, the adoption of stratified sampling would have been advisable. Additionally, the article does not specify the distribution of independent variables, which is particularly important, as these could impact the reported prevalence and correlations.

In general, the article is difficult to read due to the multiple variables involved and the way results are presented, which makes concusions unclear as well.

Additionally, recommendations for future research are missing.

Reviewer #4: This manuscript on the associations between COVID-19 conspiracy theory beliefs and sociodemographic factors, trust in government, etc. as well as with covid vaccine hesitancy tackles a relevant and complex issue.

Conceptual framework

The manuscript, however, does not include a theoretical or conceptual framework, which could significantly strengthen this manuscript.

Although not a deal breaker, it would be helpful if the authors could include a conceptual framework, perhaps with antecedents → beliefs → consequences, if not a proper theory. The inclusion of a theory or framework serves as a guiding structure for your research. Without it, the study feels somewhat exploratory or descriptive, and it becomes harder to draw clear conclusions about the mechanisms at play.

I recommend revisiting the literature review to see if there’s a theory or framework that aligns with the factors the authors are investigating. Once identified, it could frame the research questions, guide the discussion of the findings in a more structured way, and suggest avenues for future research.

Limitations

This study relies on a “nationally representative web-based self-administered questionnaire survey.” Yet, most, but certainly not all, Japanese use the Internet and the sample is self-selected. This should be mentioned in the limitations section, including any information the authors could provide regarding who is likely left out of this “nationally representative” sample.

Consequences of correlates of COVID-19 conspiracy theory beliefs

The brief description of associations between conspiracy theory beliefs and vaccine hesitancy is an important aspect of this study. I encourage the authors to give this more attention in the manuscript, including suggestions regarding future research to explore the consequences of COVID-19 conspiracy theory beliefs in addition to the correlates of such beliefs.

Note: Restrictions on data sharing are addressed in the manuscript.

6. PLOS authors have the option to publish the peer review history of their article (what does this mean?). If published, this will include your full peer review and any attached files.

Reviewer #1: No

Reviewer #2: No

Reviewer #3: **Yes: **Sara Giorgi

Reviewer #4: No

---

## [Author Response · Author response to Decision Letter 0]

22 Nov 2024

RESPONSES TO REVIEWERS’ COMMENTS

Journal requirements

and

REPLY

We have modified our manuscript files to adhere to the style requirements of PLOS ONE.

2. In the online submission form you indicate that your data is not available for proprietary reasons and have provided a contact point for accessing this data. Please note that your current contact point is a co-author on this manuscript. According to our Data Policy, the contact point must not be an author on the manuscript and must be an institutional contact, ideally not an individual. Please revise your data statement to a non-author institutional point of contact, such as a data access or ethics committee, and send this to us via return email. Please also include contact information for the third party organization, and please include the full citation of where the data can be found.

REPLY

We have modified the data statement.

Availability of data and material

The data from this study are not stored in a public repository due to the inclusion of personally identifiable or sensitive information. In accordance with Japan’s ethical guidelines, the Research Ethics Committee of the Osaka International Cancer Institute imposed restrictions on data sharing. For any inquiries regarding the data, please contact the JACSIS research office (jacsis_jastis_office@ymail.ne.jp). Details on data availability can be found on the JACSIS website (https://jacsis-study.jp/dug/index. html).

3. We note that there is identifying data in the Supporting Information file < 2_Suppl.pdf>. Due to the inclusion of these potentially identifying data, we have removed this file from your file inventory. Prior to sharing human research participant data, authors should consult with an ethics committee to ensure data are shared in accordance with participant consent and all applicable local laws.

-Location data

 Please remove or anonymize all personal information (Ages more specific than whole numbers), ensure that the data shared are in accordance with participant consent, and re-upload a fully anonymized data set. Please note that spreadsheet columns with personal information must be removed and not hidden as all hidden columns will appear in the published file.

REPLY

We carefully reviewed the website and paper you provided. These statements address dataset sharing, not tables. Our supplements contain only tables and no information that could identify individuals, even when combining table data. Therefore, our supplements are suitable for publication.

Response to Reviewer 1

I understand they have many results, but understanding can be challenging. This article should be shorter and easier to understand. I have the following comments:

REPLY

Thank you for your valuable feedback and suggestions. Please find below our point-by-point response to your comments and concerns.

1. Language: Better language (letter errors, grammar... etc.) is needed. It must use the Grammarly program to become a better manuscript.

REPLY

We have revised the manuscript using the Grammarly carefully.

2. Methodology: There are a lot of references. It must reduce references in the methodology

REPLY

We have removed some less important references in the methodology.

3. The result is a lot of mixing and results found in this manuscript. Most of the data is old, as shown: There are very long tables, and many tables are also in the appendix. It isn't easy to understand these tables.

REPLY

We've simplified the results and tables to focus on the most relevant findings. In addition, we removed some less important supplement tables. However, full tables (Tables 1, 2, and 3) are available in the supplementary file for context and transparency.

(Line 216–274, page 9–14)

Table 1 shows basic participant characteristics (full table in STable 4). The median age of respondents was 50, ranging from 16 to 81. The proportion of women was 50.8%. After imputation and weighting, 10.6% (95% confidence interval [CI] 10.1, 11.1) of the participants believed in one COVID-19 conspiracy theory (OCEANS), 6.5% (95%CI 6.1, 6.9) in two, and 7.3% (95%CI 7.0, 7.8) in three. Thus, the prevalence of harboring at least one COVID-19 conspiracy theory was 24.4% (95%CI 23.7, 25.1). The detailed results of the original categories for three questions on COVID-19 conspiracy beliefs are shown in STable 5. On the other hand, participants who believed in at least one conspiracy theory regarding general vaccines (VCBS) constituted 33.6% (detailed results are shown in STable 6).

Table 1. Basic characteristics of 28,175 participants

Variable Response participants Imputed population Weighted imputed population

　 n=28,175 (%) n=28,175 (%) n=28,175 (%)

The number of conspiracy theory beliefs regarding COVID 19

 0 21,414 (76.0%) 21,414 (76.0%) 21,297 (75.6%) 

 1  2,876 (10.2%)  2,876 (10.2%)  2,974 (10.6%) 

 2  1,830  (6.5%)  1,830  (6.5%)  1,836  (6.5%) 

 3  2,055  (7.3%)  2,055  (7.3%)  2,068  (7.3%) 

The number of conspiracy theory beliefs regarding general vaccines 

 0 18,798 (66.7%) 18,798 (66.7%) 18,712 (66.4%) 

 1  3,021 (10.7%)  3,021 (10.7%)  3,010 (10.7%) 

 2  2,022  (7.2%)  2,022  (7.2%)  2,067  (7.3%) 

 3  1,180  (4.2%)  1,180  (4.2%)  1,203  (4.3%) 

 4    873  (3.1%)    873  (3.1%)    865  (3.1%) 

 5    772  (2.7%)    772  (2.7%)    768  (2.7%) 

 6    741  (2.6%)    741  (2.6%)    779  (2.8%) 

 7    768  (2.7%)    768  (2.7%)    770  (2.7%) 

Age (years old)

 16–19    573  (2.0%)    573  (2.0%)    949  (3.4%) 

 20–24  1,829  (6.5%)  1,829  (6.5%)  1,801  (6.4%) 

 25–29  1,797  (6.4%)  1,797  (6.4%)  1,834  (6.5%) 

 30–34  1,963  (7.0%)  1,963  (7.0%)  1,885  (6.7%) 

 35–39  2,186  (7.8%)  2,186  (7.8%)  2,114  (7.5%) 

 40–44  2,600  (9.2%)  2,600  (9.2%)  2,350  (8.3%) 

 45–49  2,854 (10.1%)  2,854 (10.1%)  2,798  (9.9%) 

 50–54  2,529  (9.0%)  2,529  (9.0%)  2,661  (9.4%) 

 55–59  2,256  (8.0%)  2,256  (8.0%)  2,250  (8.0%) 

 60–64  2,234  (7.9%)  2,234  (7.9%)  2,125  (7.5%) 

 65–69  2,648  (9.4%)  2,648  (9.4%)  2,263  (8.0%) 

 70–74  2,655  (9.4%)  2,655  (9.4%)  2,781  (9.9%) 

 75–81  2,051  (7.3%)  2,051  (7.3%)  2,362  (8.4%) 

Sex

 Men 13,870 (49.2%) 13,870 (49.2%) 13,986 (49.6%) 

 Women 14,305 (50.8%) 14,305 (50.8%) 14,189 (50.4%) 

Marital status

 Married 16,969 (60.2%) 16,969 (60.2%) 17,472 (62.0%) 

 Unmarried  8,322 (29.5%)  8,322 (29.5%)  7,918 (28.1%) 

 Widowed  1,001  (3.6%)  1,001  (3.6%)  1,200  (4.3%) 

 Divorced  1,883  (6.7%)  1,883  (6.7%)  1,585  (5.6%) 

Educational attainment

 Lower Secondary School    392  (1.4%)    392  (1.4%)  1,960  (7.0%) 

 Upper Secondary School  7,592 (27.1%)  7,638 (27.1%) 12,780 (45.4%) 

 Specialised Training College (Post-Secondary Courses)  3,203 (11.4%)  3,230 (11.5%)  2,775  (9.9%) 

 Junior College and College of Technology  3,045 (10.9%)  3,065 (10.9%)  2,447  (8.7%) 

 University 12,368 (44.2%) 12,440 (44.2%)  7,421 (26.3%) 

 Master’s or doctor’s degree  1,409  (5.0%)  1,410  (5.0%)    791  (2.8%) 

 (Missing)    166 

Employment status

 Regular employee  9,722 (34.5%)  9,722 (34.5%)  9,231 (32.8%) 

 Temporary employee  4,881 (17.3%)  4,881 (17.3%)  4,976 (17.7%) 

 Self-employed  1,732  (6.1%)  1,732  (6.1%)  1,659  (5.9%) 

 Employer    990  (3.5%)    990  (3.5%)    980  (3.5%) 

 Student  1,267  (4.5%)  1,267  (4.5%)  1,387  (4.9%) 

 Unemployed or retired  4,636 (16.5%)  4,636 (16.5%)  4,768 (16.9%) 

 Home maker  4,711 (16.7%)  4,711 (16.7%)  4,938 (17.5%) 

 Others    236  (0.8%)    236  (0.8%)    235  (0.8%) 

Annual household income

 0 to <2 million yen  2,415 (10.9%)  2,575  (9.1%)  3,298 (11.7%) 

 2 to <4 million yen  5,738 (25.9%)  7,142 (25.3%)  8,298 (29.5%) 

 4 to <5 million yen  2,868 (13.0%)  3,908 (13.9%)  3,669 (13.0%) 

 5 to <8 million yen  6,082 (27.5%)  8,530 (30.3%)  7,219 (25.6%) 

 ≥8 million yen  5,024 (22.7%)  6,020 (21.4%)  5,692 (20.2%) 

 (Missing)  6,048 

Household financial assets

 0 to <1 million yen  3,034 (17.5%)  4,330 (15.4%)  5,144 (18.3%) 

 1 to <4 million yen  3,779 (21.8%)  6,707 (23.8%)  7,434 (26.4%) 

 4 to <9 million yen  3,440 (19.9%)  6,594 (23.4%)  6,245 (22.2%) 

 9 to <20 million yen  3,247 (18.7%)  5,349 (19.0%)  4,931 (17.5%) 

 ≥20 million yen  3,825 (22.1%)  5,195 (18.4%)  4,422 (15.7%) 

 (Missing) 10,850 

Household indebtedness

 None 15,722 (71.5%) 20,145 (71.5%) 19,914 (70.7%) 

 >0 to <2 million yen  1,510  (6.9%)  2,005  (7.1%)  2,236  (7.9%) 

 ≥2 million yen  4,771 (21.7%)  6,025 (21.4%)  6,025 (21.4%) 

 (Missing)  6,172 　 　 　 　 　

Full table is in STable 4.

 Table 2 presents the weighted mean of the number of COVID‑19 conspiracy beliefs (full table in STable 7). University graduates exhibited a higher endorsement of COVID‑19 conspiracy beliefs (0.49), and the highest annual household income group also showed higher conspiracy beliefs (0.52). Paradoxically, participants who distrusted the Japanese government endorsed fewer conspiracy beliefs (0.40) compared to those who trusted the government (0.56).

Table 2. Weighted means of the number of COVID-19 conspiracy beliefs after imputation

Variable Mean number of conspiracy theory beliefs regarding COVID 19

The number of conspiracy theory beliefs regarding general vaccines

 0 0.41

 1 0.48

 2 0.45

 3 0.45

 4 0.56

 5 0.50

 6 0.75

 7 1.12

Educational attainment

 Lower Secondary School 0.39

 Upper Secondary School 0.45

 Specialised Training College (Post-Secondary Courses) 0.46

 Junior College and College of Technology 0.44

 University 0.49

 Master’s or doctor’s degree 0.42

Employment status

 Regular employee 0.51

 Temporary employee 0.41

 Self-employed 0.44

 Employer 0.58

 Student 0.52

 Unemployed or retired 0.45

 Home maker 0.38

 Others 0.38

Annual household income

 0 to <2 million yen 0.44

 2 to <4 million yen 0.45

 4 to <5 million yen 0.42

 5 to <8 million yen 0.44

 ≥8 million yen 0.52

Household financial assets

 0 to <1 million yen 0.41

 1 to <4 million yen 0.48

 4 to <9 million yen 0.46

 9 to <20 million yen 0.50

 ≥20 million yen 0.42

Household indebtedness

 None 0.45

 >0 to <2 million yen 0.47

 ≥2 million yen 0.48

Information source for COVID-19: Websites of government agencies

 Not use 0.48

 Use but distrust 0.66

 Use and trust 0.41

Information source for COVID-19: Video sharing platforms (e.g. YouTube)

 Not use 0.43

 Use but distrust 0.50

 Use and trust 0.65

Information source for COVID-19: LINE

 Not use 0.43

 Use but distrust 0.57

 Use and trust 0.56

Information source for COVID-19: Twitter

 Not use 0.44

 Use but distrust 0.48

 Use and trust 0.60

Information source for COVID-19: Facebook

 Not use 0.44

 Use but distrust 0.66

 Use and trust 0.77

Information source for COVID-19: Instagram

 Not use 0.44

 Use but distrust 0.60

 Use and trust 0.74

Information source for COVID-19: Books

 Not use 0.43

 Use but distrust 0.82

 Use and trust 0.72

Information source for COVID-19: TV news

 Not use 0.54

 Use but distrust 0.48

 Use and trust 0.43

Information source for COVID-19: Tabloid TV shows

 Not use 0.46

 Use but distrust 0.43

 Use and trust 0.46

Trust in the government of Japan

 Distrust 0.40

 Trust 0.56

Full table is in STable 7.

 Table 3 shows the results of multivariable linear regression analysis with sampling weights after imputation (full table in STable 8). Lower secondary education as well as holding a master’s or doctor’s degree were associated with endorsing fewer COVID-19 conspiracy theories compared with upper secondary education (lower secondary education: B -0.089, 95%CI -0.172, -0.006; master’s or doctor’s degree: B -0.066, 95%CI -0.126, -0.006). Temporary employees (B -0.051, 95%CI -0.095, -0.007), self-employed (B -0.102, 95%CI -0.162, -0.042), unemployed or retired (B -0.078, 95%CI -0.131, -0.025), and home makers (B -0.086, 95%CI -0.135, -0.037) held fewer conspiracy theories compared to regular employees. In addition, higher household income and assets were also associated with endorsing more conspiracy theories (annual household income of ≥8 million yen: B 0.069, 95%CI 0.020, 0.117 vs. 4 to <5 million yen; household financial assets of 9 to <20 million yen: B 0.060, 95%CI 0.010, 0.109 vs. 0 to <1 million yen).

 As for information sources, participants who used and trusted the websites of government agencies had fewer conspiracy beliefs than those who did not use the websites of government (B -0.109, 95%CI -0.139, -0.079). On the other hand, participants who used and trusted video sharing platforms (e.g. YouTube) (B 0.089, 95%CI 0.038, 0.139), LINE (B 0.053, 95%CI 0.009, 0.097), Facebook (B 0.089, 95%CI 0.000, 0.178), and Instagram (B 0.093, 95%CI 0.008, 0.178) endorsed more conspiracy theories than those who did not use them. Participants who trusted the information from books endorsed more conspiracy beliefs than those who did not use books (B 0.136, 95%CI 0.068, 0.204). Participants who trusted TV news endorsed fewer conspiracy theories (B -0.081, 95%CI -0.128, -0.035 vs. not used), while those who trusted tabloid TV shows held more conspiracy theories (B 0.051, 95%CI 0.015, 0.087 vs. not used).

 Trust in authorities was associated with endorsing COVID-19 conspiracy beliefs. Trust in the government of Japan was associated with holding more conspiracy theories (B 0.175, 95%CI 0.139, 0.211), compared to those who distrusted government.

Table 3. Associations between independent variables and the number of COVID-19 conspiracy beliefs

　 Weighted multivariable adjusted model

 (n=28,175)

Variable Beta 95% confidence interval

The number of conspiracy theory beliefs regarding general vaccines

 0 (reference) — —

 1 0.059 0.017, 0.102

 2 0.045 -0.004, 0.094

 3 0.040 -0.022, 0.103

 4 0.178 0.098, 0.258

 5 0.128 0.042, 0.215

 6 0.401 0.299, 0.503

 7 0.718 0.593, 0.843

Educational attainment 

 Upper Secondary School (reference) — —

 Lower Secondary School -0.089 -0.172, -0.006

 Specialised Training College (Post-Secondary Courses) 0.018 -0.026, 0.06

---

## [Editor Report · Decision Letter 1]

10 Dec 2024

Correlates of COVID-19 conspiracy theory beliefs in Japan: A cross-sectional study of 28,175 residents

PONE-D-24-38558R1

Dear Dr. Sato,

We’re pleased to inform you that your manuscript has been judged scientifically suitable for publication and will be formally accepted for publication once it meets all outstanding technical requirements.

Kind regards,

Osmond Ekwebelem

Academic Editor

PLOS ONE
---

## [Editor Report · Acceptance letter]

16 Dec 2024

PONE-D-24-38558R1 

PLOS ONE

Dear Dr. Sato, 

I'm pleased to inform you that your manuscript has been deemed suitable for publication in PLOS ONE. Congratulations! Your manuscript is now being handed over to our production team.

Kind regards, 

on behalf of

Dr. Osmond Ekwebelem 

Academic Editor

PLOS ONE